# Non-Ventilated Patients with Spontaneous Pneumothorax or Pneumomediastinum Associated with COVID-19: Three-Year Debriefing across Five Pandemic Waves

**DOI:** 10.3390/jpm13101497

**Published:** 2023-10-15

**Authors:** Adina Maria Marza, Alexandru Cristian Cindrea, Alina Petrica, Alexandra Valentina Stanciugelu, Claudiu Barsac, Alexandra Mocanu, Roxana Critu, Mihai Octavian Botea, Cosmin Iosif Trebuian, Diana Lungeanu

**Affiliations:** 1Department of Surgery, “Victor Babes” University of Medicine and Pharmacy, 300041 Timisoara, Romania; marza.adina@umft.ro (A.M.M.); alina.petrica@umft.ro (A.P.); trebuian.cosmin@umft.ro (C.I.T.); 2Emergency Department, Emergency Clinical Municipal Hospital, 300079 Timisoara, Romania; 3Faculty of Medicine, “Victor Babes” University of Medicine and Pharmacy, 300041 Timisoara, Romania; alexandru.cindrea.umfvbt@gmail.com; 4Emergency Department, “Pius Brinzeu” Emergency Clinical County Hospital, 300736 Timisoara, Romania; 5Doctoral School, “Victor Babes” University of Medicine and Pharmacy, 300041 Timisoara, Romania; 6Clinic of Anaesthesia and Intensive Care, “Pius Brinzeu” Emergency Clinical County Hospital, 300736 Timisoara, Romania; 7Department of Infectious Diseases, “Victor Babes” University of Medicine and Pharmacy, 300041 Timisoara, Romania; alexandra.mocanu@umft.ro; 8Department of Surgery, Faculty of Medicine and Pharmacy, University of Oradea, 410087 Oradea, Romania; 9Center for Modeling Biological Systems and Data Analysis, “Victor Babes” University of Medicine and Pharmacy, 300041 Timisoara, Romania; dlungeanu@umft.ro; 10Department of Functional Sciences, “Victor Babes” University of Medicine and Pharmacy, 300041 Timisoara, Romania

**Keywords:** COVID-19, non-ventilated patients, spontaneous pneumothorax, spontaneous pneumomediastinum, complications, in-hospital mortality, emergency presentation, waves severity

## Abstract

Spontaneous pneumothorax and pneumomediastinum (SP–SPM) are relatively rare medical conditions that can occur with or independently of COVID-19. We conducted a retrospective analysis of SP–SPM cases presented to the emergency departments (EDs) of two University-affiliated tertiary hospitals from 1 March 2020 to 31 October 2022. A total of 190 patients were identified: 52 were COVID-19 cases, and 138 were non-COVID-19 cases. The primary outcome we were looking for was in-hospital mortality. The secondary outcomes concerned the disease severity assessed by (a) days of hospitalization; (b) required mechanical ventilation (MV); and (c) required intensive care (IC). All were investigated in the context of the five pandemic waves and the patients’ age and comorbidities. The pandemic waves had no significant effect on the outcomes of these patients. Logistic regression found age (OR = 1.043; 95%CI 1.002–1.085), COVID-19 (OR = 6.032; 95%CI 1.757–20.712), number of comorbidities (OR = 1.772; 95%CI 1.046–3.001), and ground-glass opacities over 50% (OR = 5.694; 95%CI 1.169–27.746) as significant risk predictors of in-hospital death while controlling for gender, smoking, the pandemic wave, and the extension of SP–SPM. The model proved good prediction performance (Nagelkerke R-square = 0.524) and would hold the same significant predictors for MV and IC.

## 1. Introduction

Spontaneous pneumothorax (SP) and spontaneous pneumomediastinum (SPM) have been frequently reported as complications associated with severe acute respiratory syndrome-coronavirus 2 (SARS-CoV-2) infection [1,2,3] and are defined as the presence of air in the pleural space or mediastinum without any history of trauma or other known cause. While the certain mechanisms for the development of these conditions in the context of coronavirus disease 2019 (COVID-19) pneumonia are not fully understood, it is believed that the virus may induce an excessive inflammatory response, making it more susceptible to the alveolar fragility and the occurrence of SP or SPM [4], denoted by SP–SPM hereafter.

Mechanical ventilation emerged as the primary contributing factor with a higher frequency of SP reported among patients with COVID-19 receiving mechanical ventilation compared to other cases of acute respiratory distress syndrome (ARDS) or viral causes [4]. Additionally, positive pressure ventilation was linked to increased mortality in intubated patients with COVID-19 [5].

The incidence of SP–SPM in patients with COVID-19 is still under research. A high occurrence of 18.4% (13–25.3%) has been reported by Shrestha et al. among patients admitted to the intensive care unit (ICU) who underwent mechanical ventilation [6]. Nevertheless, the incidence among non-ventilated patients is lower with reported rates below 1% [2,7]. A multicenter study conducted by Miro et al. that included non-ventilated patients with SP reported a low frequency of SP associated with SARS-CoV-2 infection (0.56‰). However, this frequency was still higher than in non-COVID-19 cases (0.28‰), and they had worse outcomes compared to SP in non-COVID-19 cases and sole COVID-19 cases [1].

SP–SPM in non-ventilated patients with COVID-19 is frequently diagnosed in the ED and associated with symptoms characteristic of SARS-CoV-2 infection, such as mild symptoms (e.g., fever, fatigue, and cough), severe symptoms (e.g., chest pain and dyspnea), respiratory distress, or even acute respiratory distress syndrome. Such cases would experience extended hospital stays, elevated probability of ICU admission, and increased risk of mortality, particularly among the elderly population [7,8].

During the COVID-19 pandemic, the SARS-CoV-2 virus underwent a series of mutations, resulting in multiple variants. Their virulence and level of transmissibility, and vaccine development and deployment in the population influenced COVID-19 evolution, affecting the severity of the disease, its clinical presentations, and mortality.

Palumbo et al. conducted the first study that compared the frequency and characteristics of SP–SPM in non-ventilated patients hospitalized with SARS-CoV-2 infection during the first two pandemic waves. They identified 14 patients with SP–SPM, reporting a 29% admission rate to the ICU and a 50% mortality rate. The study also revealed a significantly higher frequency of this complication during the second wave of the pandemic. However, it is important to note that the study included only two waves, which resulted in a limited number of patients, particularly from the first wave where they observed only one patient [9]. Furthermore, Tacconi et al. also reported a higher frequency of pneumomediastinum during the second wave compared to the first one, but their study also included mechanically ventilated patients in both waves [10].

We conducted a retrospective analysis of the SP–SPM cases presented between 1 March 2020 and 31 October 2022 to the emergency departments (EDs) of Emergency Clinical Municipal Hospital and “Pius Brinzeu” Emergency Clinical County Hospital, two tertiary hospitals affiliated with “Victor Babes” University of Medicine and Pharmacy from Timisoara.

The main objective of our research was to investigate the risk of in-hospital mortality in ED patients with SP–SPM during the COVID-19 pandemic. Specifically, we sought to observe a hypothesized relation of an increased risk of mortality with the SARS-CoV-2 infection and any possible patterns across the five pandemic waves. The secondary objectives concerned the SP–SPM disease severity assessed by (a) days of hospitalization; (b) required mechanical ventilation; and (c) required intensive care. All three secondary outcomes were also considered in the context of the five pandemic waves.

## 2. Materials and Methods

### 2.1. Study Design and Patients

A retrospective analysis of the electronic medical records (EMRs) was conducted, and 206,097 patients who presented at EDs were retrieved. The inclusion criteria were age above 18 years, presence of SP–SPM confirmed with chest X-ray or computed tomography (CT), and rapid antigenic testing and/or reverse transcription–polymerase chain reaction (RT–PCR) test for SARS-CoV-2 infection. The exclusion criteria were age below 18 years, post-traumatic SP–SPM, and iatrogenic or post-mechanical ventilation SP–SPM.

A keyword-based automated search in the hospital computer system identified 743 cases of SP–SPM. The diagnosis was determined based on the medical records at the time of hospital discharge and defined in accordance with the International Classification of Diseases, 10th Revision, Clinical Modification (ICD-10-CM). The diagnosis code U07.1 was used to identify cases of COVID-19, while J93.11 (SP) or J98.2 (SPM) were employed to designate cases of SP–SPM. Several cases were excluded as they did not meet the study protocol criteria: six patients were under the age of 18 years, 12 patients had iatrogenic (post-procedural) SP, 508 patients had post-traumatic SP–SPM, and 27 cases were associated with mechanical ventilation (e.g., following cardiopulmonary resuscitation). The remaining 190 cases of SP–SPM were manually reviewed to ensure compliance with the inclusion criteria and completeness of the relevant medical data. Figure 1 presents the study flow diagram.

Public data regarding the daily number of COVID-19 cases were retrieved from the Centre for Transmissible Disease Control of Timisoara City Council website [11]. The time spells corresponding to the pandemic waves were determined based on the reports from the National Institute of Public Health [12] and corroborated with the results from a recently published study conducted in “Victor Babes” Clinical Hospital for Infectious Diseases and Pulmonology in Timisoara [13]. We were, thus, able to document the wide range of cumulative cases over the five pandemic waves. Nevertheless, it should be noted that, during the first two waves, there were no standard protocols for the management of patients suspected or confirmed to have COVID-19. Table 1 and Figure 2 show the details of the five pandemic waves.

From March 2020, rapid antigenic testing and/or reverse transcription-polymerase chain reaction (RT-PCR) for SARS-CoV-2, along with chest imaging (X-ray or chest computed tomography, depending on the patient’s diagnosis) were mandatory, according to the hospitals’ internal protocols, regardless of the symptoms of the admitted patients (either on the ward or ICU) through the ED.

In Romania, the initial wave did not have a significant increase in the number of cases due to the authorities’ restrictions, leading to a remarkably low count of infections. As a result, no spike could be identified during the first wave. There was not any distinct demarcation between the second and third waves, as the spikes in these two waves were very close in terms of timing (Figure 2). This could be explained by the lack of compliance with the restrictions imposed by the authorities during that period and the growing public mistrust of the healthcare system [14]. The patients infected between the two waves were categorized as belonging to the preceding wave, while those infected after the fifth wave were classified as part of that wave.

### 2.2. Data Collection

The EMR retrieved data comprised demographics, date of admission to the ED, patient medical history (symptoms, SpO2 value upon arrival in ED, comorbidities including smoking status, history of SARS-CoV-2 infection), paraclinical investigations (laboratory tests, radiological findings on X-ray or chest-CT), decided treatments (observation or chest tube), hospitalization days, ICU admission, need for mechanical ventilation (in hospital, after SP–SPM diagnosis), and discharge outcome. All records were de-identified.

For the present analysis, the laboratory tests on the first day of presentation in the ED (the first 24 h) were taken into consideration; some patients had additional blood tests (e.g., C-reactive protein, procalcitonin) collected at the time of ward admission; some patients stayed for several days in the ED (due to a shortage of beds in COVID-19 support hospitals), and therefore, several sets of laboratory tests were processed.

### 2.3. Data Analysis

The normality of the data was tested using the Shapiro–Wilk test. The descriptive statistics for the categorical variables were frequency counts and corresponding percentages. The numerical variables were described by the mean with standard deviation, irrespective of their distribution. The rank variables with less than four values were treated as categorical. The statistical significance of the associations between the categorical variables was tested with the chi-square test (either asymptotic or Monte Carlo simulation with 10,000 samples). The significance of the numerical variables’ distributions across the pandemic waves was tested with the nonparametric Kruskal–Wallis statistical test, separately for COVID-19 positive and negative cases.

Logistic regression was employed to identify the significant independent predictors of the binary outcomes, namely in-hospital death, required mechanical ventilation, and required intensive care. The medically meaningful predictors that were statistically significant in the univariate analysis were progressively introduced, and the candidate regression models were compared based on the Akaike information criterion (AIC) and Nagelkerke R-square coefficient.

The reported probability values were two-tailed, and the statistical significance was conducted at a 5% level (95% level of confidence). The data analysis was conducted with the statistical software IBM SPSS v. 20 and R v. 4.3.1 packages.

### 2.4. Ethics

This study was conducted in accordance with the Declaration of Helsinki, and the protocol was approved by the Ethics Committees of the Emergency Clinical Municipal Hospital Timisoara (number I-1831/31 March 2023) and “Pius Brinzeu” Emergency Clinical County Hospital Timisoara (number 387/22 March 2023). The collected data were de-identified before conducting the statistical analysis. Considering the retrospective nature of the study as well as the importance of obtaining new important data regarding the consequences of the novel coronavirus infection, the patient’s informed consent was waived.

## 3. Results

### 3.1. Characteristics of Patients with SP–SPM

Out of all presentations (206,097 patients) for any cause in the two emergency departments, the patients with pneumothorax and/or pneumomediastinum accounted for 0.03%, while SP–SPM represented 0.009% of the cases. A total of 190 eligible SP–SPM cases were identified: 52 were positive, and 138 were negative for SARS-CoV-2 infection. Seven patients denied hospitalization (four in Wave 5, two in Wave 4, and one in Wave 3). Out of these, three patients had both SP and SPM (one confirmed with COVID-19), one patient had isolated SPM, and three patients had isolated SP (one confirmed with COVID-19). Only one patient experienced an extensive pneumothorax (exceeding 50% of the lung), while the remaining cases had a pneumothorax extent below 10%.

The male gender was predominant (72.63%). The prevailing comorbidities among all patients in the study included hypertension as the most frequent, followed by COPD and diabetes mellitus. In Wave 5, a higher number of cancer cases were registered. Table 2 synthesizes the patients’ general characteristics.

Table 3 and Table 4 show the main radiological findings encountered in the patients with SP–SPM, the treatment administered in the ED, as well as the extent of GGO involvement in the patients with COVID-19 with SP–SPM. The most common imagistic findings were GGOs, lung infiltrates, and pneumomediastinum, both in the COVID-19 and non-COVID-19 cases. Chest tube insertion as an emergency treatment for pneumothorax was more frequent during the early waves of the pandemic. The majority of the patients with COVID-19 with GGOs exceeding 50% were observed during the third wave, aligning with the decreased oxygen saturation levels in that same period. The detailed descriptive statistics for the medical information retrieved from the patients’ records are presented in the Appendix A.

Across these five waves, we observed 11 non-COVID-19 cases with SP–SPM that presented with a history of SARS-CoV-2 infection. Out of these patients, two were suffering from SPM alone and one from both SP and SPM. One patient died due to a massive SP after two days of hospitalization while being admitted to the ICU and after he had a chest tube inserted. None of these patients had significant comorbidities, 64% were males, their average age was 60 years (24–88), and their clinical feature was dominated by cough and chest pain. Six of them had the C-reactive protein checked; out of these, five had elevated values (average 121 mg/L with a 277 mg/L uppermost value). In addition, there were five patients with persistent ground-glass opacities: three of them exceeded 50% of the pulmonary surface, and two were between 10% and 50%.

Table 5 synthesizes the outcomes related to the study objectives. Figure 3 shows the box-plot diagram of the length of hospitalization across the five waves. There are many outliers and extreme values (open bullets and stars, respectively) in all waves, but there is a particularly high extreme value in Wave 2 for a COVID-19 case. The lack of a distinctive pattern is easily observable.

### 3.2. Logistic Regression Models of the Binary Outcomes

The pandemic waves had no significant effect on the risk of death. Table 6 summarizes the results of the logistic regression analysis related to in-hospital mortality as the outcome: the mortality was highly dependent on the presence of COVID-19, the increased number of comorbidities, and the presence of ground-glass opacities over 50%. The odds ratio (OR) values of the significant predictors and their corresponding confidence intervals on a logarithmic scale are depicted in Figure 4. Although some of these intervals are large (such as those for COVID-19 condition and the presence of ground-glass opacities), implying imprecision, the Nagelkerke R-square value (i.e., R-square = 0.524) demonstrates the model’s good prediction performance: these four predictors would explain more than 50% of the SP–SPM mortality when controlling for gender, smoking habits, pandemic wave, and extension of SP–SPM. On the other hand, the OR value for age is very close to 1 with a highly tight confidence interval, thus implying little impact on the outcome (statistically significant nonetheless).

Table 7 and Table 8 present the results of the logistic regression analyses with required intensive care and required mechanical ventilation as the outcomes, respectively. In both models, age was non-significant, and the increase in the number of comorbidities raised the risk of an unfavorable outcome. The presence of COVID-19 and extensive ground-glass opacities were both significant predictors with a high impact on the outcomes in each of these two risk models. Their different relative influence on the outcomes should be noted in Figure 5: COVID-19 had a significant impact on the ICU risk and halved the odds for mechanical ventilation, while extended ground-glass opacities were found to increase the odds of mechanical ventilation almost seven times and close to five times for the ICU. The Nagelkerke R-square values are smaller for these two models (0.419 and 0.373, respectively), implying that there are additional predictors and covariates influencing the outcomes.

## 4. Discussion

Each pandemic wave presented unique challenges both to the medical professionals involved in treating the patients with COVID-19 and to the healthcare system itself, thus entailing continuous research aimed at optimizing patient care. Our retrospective analysis investigated SP–SPM as a medical condition in the five waves of the COVID-19 pandemic between 1 March 2020 and 31 October 2022. The objective of this debriefing was to identify possible patterns of disease and the risk factors associated with in-hospital mortality (primary outcome) and high disease severity (secondary outcomes). Although the patients with COVID-19 were at a significantly higher risk, there were no distinct patterns across the pandemic waves.

In December 2021, the Omicron variant became the predominant lineage of SARS-CoV-2 in several European countries and was associated with significantly lower clinical severity, reduced oxygen requirements, lower rates of hospitalization, and decreased mortality rates [15]. Amodio et al. presented evidence indicating a general decrease in viral virulence from the wild-type to Omicron variants and a higher risk of MV and severe outcomes for the corresponding Delta and wild-type waves [16]. Our findings were similar regarding the hospitalization length for patients with SP–SPM with SARS-CoV-2 infection, which was longer during the second and third waves compared to that of the non-COVID cases. During the second wave, the increase in the average hospitalization was partially attributed to a single patient who required hospitalization for 140 days as a result of septic complications that arose from both COVID-19 pneumonia and the chest tube insertion. However, the hospital stay would slowly decrease in subsequent waves.

In our investigation, most of the SP–SPM cases associated with SARS-CoV-2 infection (55.9% of the total number) were identified during the second wave. These patients presented with the lowest SpO2 levels upon their arrival in the emergency department and had the longest stay in the hospital, prolonged MV (due to extensive lung damage), and a higher admission rate to the ICU. Unfortunately, they also experienced a significant mortality rate of 29.2%. These patients were at risk of developing ventilator-associated pneumonia and other nosocomial infections due to their extended hospitalization, which could have contributed to the observed increase in mortality. A systematic review and meta-analysis conducted by Shrestha et al. reported an increased occurrence of barotrauma in patients with severe forms of the disease, implying a direct connection between barotrauma and disease severity [6]. Regarding the virulence of the virus corresponding to each wave, a study conducted in 2021 by Zawbaa et al. examined the severity of the SARS-CoV-2 virus in 12 countries and revealed that the third and fourth waves were associated with higher mortality [17].

### 4.1. Mortality and Severity of Disease across the Five Pandemic Waves

In our data, the logistic regression analysis found that the presence of COVID-19 increased the in-hospital mortality risk more than six times (namely with an OR = 6.032) for patients with SP–SPM of the same age, gender, smoking status, pandemic wave, and severity of GGOs and comorbidities. Moreover, GGOs exceeding 50% increased the risk of death by approximately six times (OR = 5.694), and an increased number of comorbidities almost doubled the risk (OR = 1.772). In contrast, the pandemic wave had no significant effect on the risk of death in patients with SP–SPM. As other studies reported, the Omicron variant had substantially increased transmissibility and resulted in higher rates of prolonged hospitalization and mortality in unvaccinated patients [15]. Considering that Romania had a consistently low vaccination rate, this could explain the increased mortality in the subsequent waves despite the less aggressive virus strains. However, it is important to also acknowledge the role of natural immunity and the fact that better outcomes in some healthcare systems could be attributed to the improved clinical and therapeutical practices that resulted from increased knowledge and experience over time [16].

In a recent study, Akyil et al. reported no mortality difference in patients with SP–SPM with COVID-19 when compared to non-COVID-19 cases. However, it is also important to note that, in their study, the patients with COVID-19 were asymptomatic and did not have underlying lung lesions or GGOs. The only notable difference was a prolonged recovery time in the COVID-19 group [18].

Starting from the second wave, the proportion of SP–SPM cases associated with COVID-19 diminished, but it surged in the fifth wave, which was characterized by the Omicron variant. This pattern may suggest a significant susceptibility to SP or SPM among individuals who either had or might have experienced a prior SARS-CoV-2 infection, considering that this complication has already been described in several studies [19,20,21,22]. Despite all that, the PCR test for detecting SARS-CoV-2 infection may yield false-negative results if it is performed too early or too late in the course of the disease. This is especially relevant since it was frequently performed as a mandatory test upon admission, even in cases where patients did not exhibit COVID-19 symptoms.

### 4.2. Medical History and Comorbities of Patients with SP–SPM

In our study, the mean age of the entire group of patients with SP–SPM was 52.8 ± 18.5 years, and it was higher among those with COVID-19 (63.37 ± 13.9 years), particularly during the third wave of the pandemic (71 ± 13.5 years). This might explain the high mortality rate observed during this wave in the patients with SP–SPM with associated SARS-CoV-2 infection, considering that elder age was frequently associated with increased mortality [1,3]. In our analysis concerning in-hospital mortality, age had the least influence on the outcome. Additionally, in the secondary outcomes’ investigation, age was no longer significant after accounting for the risks associated with mechanical ventilation and ICU admission.

Most of the patients who contracted SARS-CoV-2 and subsequently developed SP–SPM were males and non-smokers, aligning with the findings in the current literature [7], where smoking was not identified as a risk factor for SP in patients with COVID-19 but was associated with greater disease severity. Regardless of the virus variant’s severity or of the pulmonary damage, the patients’ comorbidities directly influenced mortality, as evidenced in our study. For instance, in the patients infected with SARS-CoV-2 during the fourth wave of the COVID-19 pandemic (with the Delta strain), ischemic heart disease was associated with greater disease severity and increased mortality rates, surpassing those observed in the earlier waves [23].

The most frequent comorbidities among all patients included hypertension, followed by COPD and diabetes mellitus, as observed in Appendix A. Hypertension exhibited a statistically significant higher occurrence among patients with COVID-19 across all pandemic waves, which can be explained by the older age of these patients compared to the non-COVID-19 ones. Regarding pre-existing lung disease, Chong et al. reported a prevalence of less than 30% among patients with SP–SPM and COVID-19 [7]. While COPD is frequently observed in patients with SP–SPM, our dataset showed no association with the COVID-19 condition, unlike asthma (which appears to be a common comorbidity in patients with COVID-19, particularly during the third wave of the pandemic). Wang et al. [24] reported no substantial disparity between mortality rates in patients with COVID-19 with asthma and those without. Regarding the requirement for ICU admission or mechanical ventilation in patients with COVID-19 with asthma, the existing literature does not identify any discernible distinctions [25]. Our study findings may suggest that underlying pulmonary diseases alone would not significantly increase mortality in patients with SP–SPM. On the contrary, it appeared that numerous comorbidities would exacerbate the SARS-CoV-2 infection and the associated inflammatory syndrome and, thus, play an important role in the mortality increase.

In our investigation, dyspnea emerged as the most frequent symptom, followed by fatigue and cough (Appendix A). The mechanism of SP in patients with COVID-19 is far from being fully understood. Multiple hypotheses have been proposed, including delayed alveolar breach as part of a chronic inflammatory process [26] or ischemic breakdown of the alveolar wall secondary to micro-thrombi [22]. Considering that, in several studies, cough was described as a frequent symptom of SARS-CoV-2 infection and post COVID-19 syndrome [27], its increased occurrence in our dataset could be a warning regarding the patients with COVID-19. Also, dyspnea and increased respiratory drive due to SARS-CoV-2 infection might cause patient self-inflicted lung injury (P-SILI) [28] and could also be incriminated in the development of SP–SPM.

### 4.3. Imaging and Laboratory Findings in Patients with SP–SPM

The most common radiographic finding (as described in Appendix A) was right-sided pneumothorax, observed in 57.89% of patients, which is consistent with the results of other studies [1,3,7].

In the second pandemic wave, GGOs were observed in approximately all COVID-19 cases with the largest extent of these lesions (50% of patients exhibited GGOs covering > 50% of the lung fields). A high incidence of GGO lesions also persisted in the third wave, followed by a gradual reduction towards the fifth wave, wherein both the incidence and the extent of the lesions were reduced by half. Nevertheless, the mortality rate in the fifth wave remained high among the patients with COVID-19 compared to the non-COVID-19 cases (20.8% vs. 14.3%, respectively). This suggests that there were other contributory factors to the mortality risk in the individuals with SP–SPM beyond the extent of GGOs (which would increase the severity of COVID-19 pneumonia). One of them was found to be the presence of several comorbidities in our logistic regression analysis. Additionally, the large number of infections and subsequent overcrowding in COVID-19 wards, reduced rate of medical staff to treat patients, and nosocomial infections associated with prolonged hospitalization and mechanical ventilation could explain the persistence of increased mortality in the fifth wave.

Across the pandemic waves, the frequency of GGO lesions on chest CT scans increased among the non-COVID cases with SP–SPM (reaching from 0% in the first wave to 50% in the fifth wave). This trend might imply a potential history of SARS-CoV-2 infection in these cases and would indicate a higher risk of SP or SPM in patients with such a medical record, as was mentioned in several reports [22,29,30,31].

Among the patients with SP–SPM with COVID-19, the laboratory findings indicated a greater incidence of lymphopenia and increased levels of C-reactive protein, urea, and creatinine compared to the non-COVID-19 cases (Appendix A). These values were found as excessive throughout the pandemic waves, but they were most likely related to viral involvement.

### 4.4. Treatment Options in Patients with SP–SPM

The treatment of choice for the non-COVID-19 cases was chest tube, opposite to the COVID-19 group where the treatment of choice was conservative. We noted a slight reduction in the number of chest tubes used in the second part of the pandemic (in the fourth and fifth waves) with an increased number of patients for whom observation was decided as the most appropriate course of action. This approach can be explained by both the increased experience regarding the treatment of SP–SPM cases associated with SARS-CoV-2 infection and the recent pneumothorax management guideline that recommended a less invasive approach; according to the latest British Thoracic Society Guideline for pleural disease [32], the size of the pneumothorax is no longer a criterion for inserting a chest drain, which is now recommended for high-risk patients only.

### 4.5. Limitations of This Debriefing

Our study had several limitations, primarily stemming from its retrospective design. First, the absence of standardized protocols resulted in a non-uniform management of these cases, particularly during the initial waves of the pandemic. Consequently, crucial inflammatory markers in COVID-19, such as interleukin-6 and ferritin, could not be consistently determined in the ED due to a lack of reagents. CRP, procalcitonin, and D-dimers were only collected upon specific requests from the physicians, making it impossible for us to correlate the inflammatory syndrome with the occurrence of SP–SPM. Second, our hospitals did not allow routine phenotyping for each patient with COVID-19. The implementation of such phenotyping would have incurred substantial costs for an already overwhelmed healthcare system. Therefore, we considered the most prevalent national variant as the involved strain. Third, there is a possibility that some cases went undiagnosed. Some patients who presented to the hospital during the pandemic with suspected SP–SPM and tested positive for SARS-CoV-2 in the ED refused further imaging investigations and hospitalization due to distrust in the medical system and were not included in our study. Fourth, there might have been false-negative SARS-CoV-2 test results. Some patients could have been tested too early or too late in the course of their illness, leading to their erroneous inclusion in the non-COVID group of this analysis.

Despite its limitations, this analysis brought reliable evidence of no differences in mortality among the patients with SP–SPM according to the pandemic wave or severity of the virus strain. Since the emergence of the Omicron variant, severe forms of COVID-19 pneumonia have become less common. Nevertheless, we must remain vigilant regarding the heightened risk of mortality in these patients, irrespective of age. Moreover, frequent and vigilant monitoring of patients with COVID-19 should be performed in order to promptly detect the occurrence of SP. Lung ultrasound can serve as a valuable diagnostic tool at the bedside for patients with COVID-19 admitted to the ward. This approach can help avoid non-invasive ventilation or invasive mechanical ventilation in patients with SP and the associated risk of tension pneumothorax. Additionally, patients with SP–SPM and GGOs covering more than 50% of the lung surface should be considered for admission to the ICU or an intermediate care unit, depending on the hospital’s policies. This level of care can ensure continuous monitoring and surveillance, which is crucial given the increased mortality risk associated with this predictor.

## 5. Conclusions

Over approximately three years, the risks of mortality, mechanical ventilation, and ICU admission in patients with SP–SPM were significantly dependent on the coexisting COVID-19 disease, extensive lung injury, and increased number of comorbidities, regardless of the age of the patients and the virulence of the different strains involved in a pandemic wave. Beyond the COVID-19 pandemic, these findings highlight the importance of careful observation of each SP–SPM case, accompanied by personalized and rapid intervention.

## Figures and Tables

**Figure 1 jpm-13-01497-f001:**
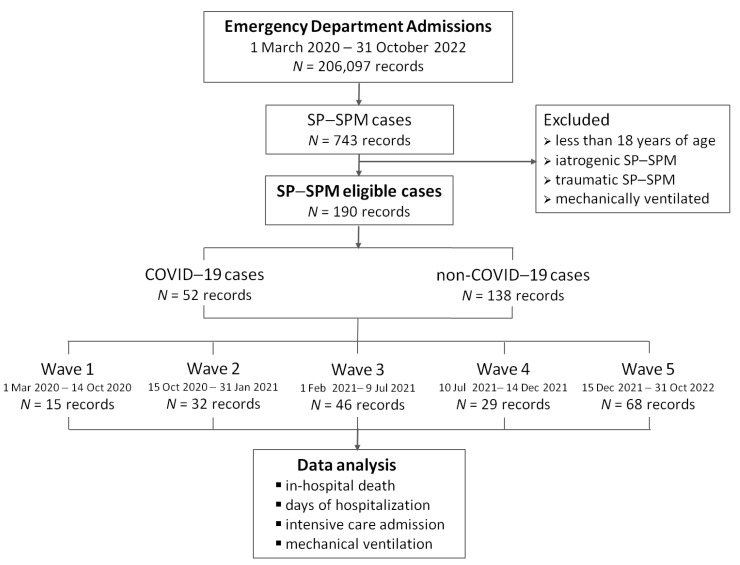
Study flow diagram. SP–SPM stands for spontaneous pneumothorax or spontaneous pneumomediastinum.

**Figure 2 jpm-13-01497-f002:**
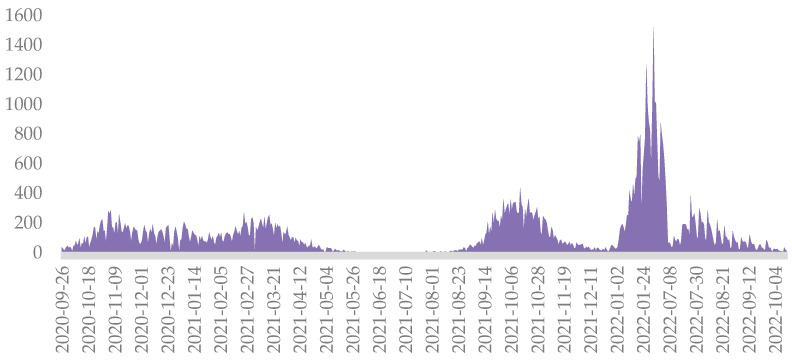
Daily case count in Timisoara from September 2020 to October 2022.

**Figure 3 jpm-13-01497-f003:**
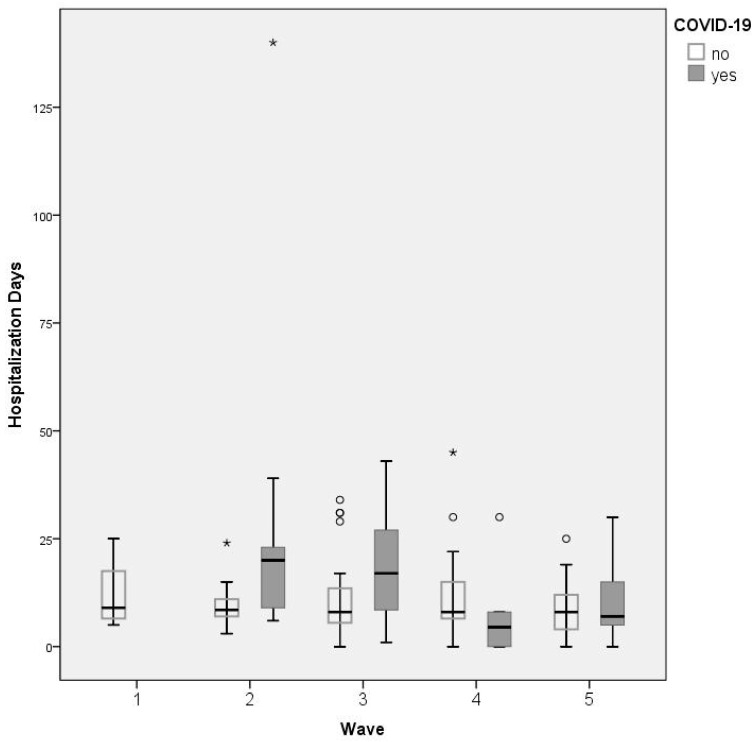
Box-plot diagram with length of hospitalization for COVID-19 and non-COVID-19 cases. The boxes are proportional to the interquartile range (IQR) with medians marked in-between, and the whiskers are proportional to 1.5 * IQR. (or trimmed to the minimum or maximum values).

**Figure 4 jpm-13-01497-f004:**
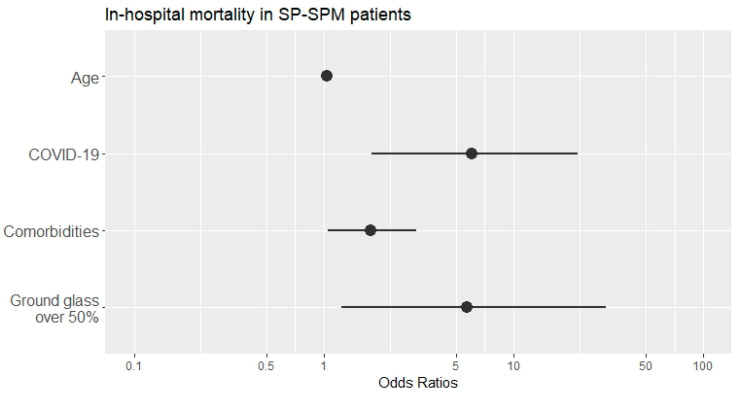
The independent predictors for in-hospital mortality as resulted from the logistic regression analysis. The dots correspond to the estimated OR values (shown in Table 6), and the lines depict the 95%CI intervals. The horizontal axis employs a logarithmic scale.

**Figure 5 jpm-13-01497-f005:**
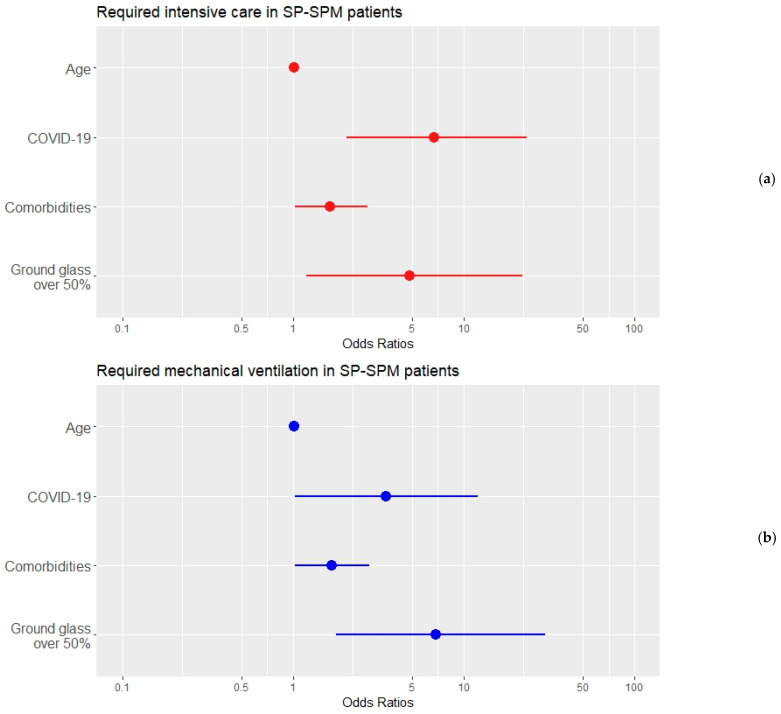
The independent predictors for required intensive care (**a**) and mechanical ventilation (**b**) in patients with SP–SPM as resulted from the logistic regression analysis. The dots correspond to the estimated OR values (shown in Table 7 and Table 8, respectively), and the lines depict the 95%CI intervals. The horizontal axes employ a logarithmic scale.

**Table 1 jpm-13-01497-t001:** The time periods corresponding to the pandemic waves in Timisoara, the prominent variants of concern involved, and the number of SP–SPM cases.

	Wave 1	Wave 2	Wave 3	Wave 4	Wave 5	
Wave period	1 Mar 2020–14 Oct 2020	15 Oct 2020–31 Jan 2021	1 Feb 2021–09 Jul 2021	10 Jul 2021–14 Dec 2021	15 Dec 2021–31 Oct 2022	
Wave spike(s)	–	22 Oct 2020–4 Jan 2021	16 Feb 2021–7 Apr 2021	15 Sep 2021–14 Nov 2021	5 Jan 2022–13 Feb 2022	19 Jul 2022–13 Sep 2022
Spike duration (in days)	–	74	50	60	39	56
Frequent VOC	Wuhan-Hu-1 NC_045512.2	Clade S:D614G	Alpha B.1.1.7	Delta B.1.617.2	Omicron B.1.1.529	
Daily average case count during the spike	–	143.16	164.64	236.71	608.61	137.32
SP–SPM cases	15	32	46	29	68	
SP–SPM and COVID-19 cases	–	18 (55.9%)	15 (31.1%)	6 (20.7%)	13 (19.1%)	

Abbreviations: VOC, variant of concern; SP–SPM, spontaneous pneumothorax and/or spontaneous pneumomediastinum.

**Table 2 jpm-13-01497-t002:** Descriptive statistics for demographics and number of most frequent comorbidities for each pandemic wave.

Variable			All Patients (N = 190)	Wave 1 (N = 15)	Wave 2 (N = 32)	Wave 3 (N = 46)	Wave 4 (N = 29)	Wave 5 (N = 68)	*p*-Value ^(a),(b)^
Age ^(a)^		+ COVID-19	63.37 ± 13.9	-	60.4 ± 12.56	71 ± 13.5	51.83 ± 9.58	63.92 ± 13.96	0.019 *
		− COVID-19	48.83 ± 18.51	46.67 ± 16.41	53.64 ± 12.51	44.45 ± 19.27	46.13 ± 17.19	51.78 ± 20.12	0.314
Gender (male) ^(b)^		+ COVID-19	40 (21%)	-	14 (35%)	9 (22.5%)	5 (12.5%)	12 (30%)	<0.001 **
		− COVID-19	98 (51.57%)	13 (13.3%)	7 (7.1%)	23 (23.5%)	14 (14.3%)	41 (41.8%)	
Active smoker ^(b)^		+ COVID-19	10 (5.26%)	-	3 (30%)	3 (30%%)	1 (10%)	3 (30%)	<0.001 **
		− COVID-19	59 (31.05%)	12 (20.3%)	8 (13.6%)	14 (23.7%)	5 (8.5%)	20 (33.9%)	
Comorbidities ^(b)^									
	none	+ COVID-19	19 (21.3%)	-	5 (26.3%)	4 (21.1%)	5 (26.3%)	5 (26.3%)	0.043 *
		− COVID-19	70 (78.7%)	7 (10%)	4 (5.7%)	19 (27.1%)	13 (18.6%)	27 (38.6%)	
	one	+ COVID-19	10 (22.2%)	-	5 (50%)	1 (10%)	-	4 (40%)	
		− COVID-19	35 (77.8%)	3 (8.6%)	3 (8.6%)	6 (17.1%)	8 (22.9%)	15 (42.9%)	
	two	+ COVID-19	12 (36.4%)	-	7 (58.3%)	3 (25%)	-	2 (16.7%)	
		− COVID-19	21 (63.6%)	2 (9.5%)	4 (19%)	3 (14.3%)	2 (9.5%)	10 (47.6%)	
	three or more	+ COVID-19	11 (47.8%)	-	1 (9.1%)	7 (63.6%)	1 (9.1%)	2 (18.2%)	
		− COVID-19	12 (52.2%)	3 (25%)	3 (25%)	3 (25%)	-	3 (25%)	

^(a)^ Mean ± SD, Kruskal–Wallis nonparametric test (separate tests for COVID-19 positive and negative patients). ^(b)^ Observed frequency (percent); chi-square test. Calculation of percentages: “All patients” column shows percentages based on the total number of patients presenting each symptom or comorbidity; all other columns show percentages based on the row total. *, ** Statistical significance, *p* < 0.05, *p* < 0.01.

**Table 3 jpm-13-01497-t003:** Radiological findings (computed tomography and/or chest radiography) and preferred treatment during each successive wave of the pandemic.

Variable ^(a)^		All Patients (N = 190)	Wave 1 (N = 15)	Wave 2 (N = 32)	Wave 3 (N = 46)	Wave 4 (N = 29)	Wave 5 (N = 68)	*p*-Value ^(a)^
Lung infiltrates	+ COVID-19	20 (71.4%)	-	8 (40%)	8 (40%)	1 (5%)	3 (15%)	<0.001 ** ^(a)^
	− COVID-19	8 (28.6%)	1 (12.5%)	2 (25%)	3 (37.5%)	1 (12.5%)	1 (12.5%)	
Ground-glass opacities	+ COVID-19	32 (84.2%)	-	16 (50%)	10 (31.3%)	2 (6.3%)	4 (12.5%)	<0.001 ** ^(a)^
	− COVID-19	6 (15.8%)	-	1 (16.7%)	1 (16.7%)	1 (16.7%)	3 (50%)	
Pneumomediastinum	+ COVID-19	23 (60.5%)	-	9 (39.1%)	8 (34.8%)	1 (4.3%)	5 (21.7%)	<0.001 ** ^(a)^
	− COVID-19	15 (39.5%)	-	1 (6.7%)	4 (26.7%)	3 (20%)	7 (46.7%)	
Subcutaneous emphysema	+ COVID-19	16 (53.3%)	-	7 (43.8%)	6 (37.5%)	1 (6.3%)	2 (12.5%)	<0.001 ** ^(a)^
	− COVID-19	14 (46.7%)	1 (7.1%)	2 (14.3%)	-	4 (28.6%)	7 (50%)	
Observation	+ COVID-19	22 (64.7%)	-	7 (30.4%)	6 (26.1%)	2 (8.7%)	8 (34.8%)	<0.001 ** ^(b)^
	− COVID-19	14 (37.8%)	-	-	3 (21.4%)	2 (14.3%)	9 (64.3%)	
Chest tube	+ COVID-19	29 (19%)	-	11 (37.9%)	9 (31%)	4 (13.8%)	5 (17.2%)	
	− COVID-19	124 (81%)	15 (12.1%)	14 (11.3%)	28 (22.6%)	21 (16.9%)	46 (37.1%)	

^(a)^ Observed frequency (percent); chi-square test. Calculation of percentages: “All patients” column shows percentages based on the total number of patients presenting each symptom or comorbidity; all other columns show percentages based on the row total. ^(b)^ Mean ± SD. ** Statistical significance, *p* < 0.01.

**Table 4 jpm-13-01497-t004:** Extension of ground-glass opacities and SpO2 values for the patients with COVID-19.

Variable		All Patients N = 52 ^(#)^	Wave 2 N = 18	Wave 3 N = 15	Wave 4 N = 6 ^(#)^	Wave 5 N = 13	*p*-Value ^(a),(b)^
GGO ^(a)^							0.035 *
	no GGO	17 (33.3%)	2 (11.8%)	5 (29.4%)	2 (11.8%)	8 (47.1%)	
	GGO < 20%	4 (7.8%)	3 (75%)	-	1 (25%)	-	
	GGO 20–50%	10 (19.6%)	4 (40%)	4 (40%)	-	2 (20%%)	
	GGO > 50%	20 (39.2%)	9 (45%)	6 (30%)	2 (10%)	3 (15%)	
SpO2 on room air ^(b)^		80.55 ± 13.81	75.72 ± 16.37	82.27 ± 10.96	84.67 ± 14.81	83.58 ± 11.75	0.363

^(a)^ Observed frequency (percent); chi-square test. Calculation of percentages: ”All patients” column shows percentages based on the total number of patients presenting each symptom or comorbidity; all other columns show percentages based on the row total. ^(b)^ Mean ± SD, Kruskal–Wallis nonparametric test. * Statistical significance, *p* < 0.05. ^(#)^ One patient refused to be admitted to the hospital and denied CT for further evaluation. Abbreviations: CT, computed tomography; GGO, ground-glass opacity; SD, standard deviation; SpO2, peripheral oxygen saturation at hospital admittance.

**Table 5 jpm-13-01497-t005:** Outcomes for the COVID-19 and non-COVID-19 cases across the five pandemic waves.

Variable		All Patients (N = 190)	Wave 1 (N = 15)	Wave 2 (N = 32)	Wave 3 (N = 46)	Wave 4 (N = 29)	Wave 5 (N = 68)	*p*-Value ^(a),(b)^
Deceased ^(a)^	+ COVID-19	24 (77.4%)	-	7 (29.2%)	9 (37.5%)	3 (12.5%)	5 (20.8%)	<0.001 **
	− COVID-19	7 (22.6%)	-	2 (28.6%)	4 (57.1%)	-	1 (14.3%)	
Hospitalization days ^(b)^	+ COVID-19	17.63 ± 20.82	-	24.72 ± 30.38	18.27 ± 12.58	7.83 ± 11.39	11.08 ± 10.04	0.035 *
	− COVID-19	9.88 ± 7.58	12.07 ± 7	9.5 ± 5.19	10.81 ± 9.09	12 ± 10.02	7.94 ± 5.62	0.285
Required ICU ^(a)^	+ COVID-19	23 (71.9%)	-	9 (39.1%)	6 (26.1%)	2 (8.7%)	6 (26.1%)	<0.001 **
	− COVID-19	9 (28.1%)	-	2 (22.2%)	2 (22.2%)	2 (22.2%)	3 (33.3%)	
Required MV ^(a)^	+ COVID-19	19 (65.5%)	-	6 (31.6%)	5 (26.3%)	2 (10.5%)	6 (31.6%)	<0.001 **
	− COVID-19	10 (34.5%)	-	2 (20%)	2 (20%)	3 (30%)	3 (30%)	

^(a)^ Observed frequency (percent); chi-square test. Calculation of percentages: ”All patients” column shows percentages based on the total number of patients presenting each symptom or comorbidity; all other columns show percentages based on the row total. ^(b)^ Mean ± SD, Kruskal–Wallis nonparametric test. *, ** Statistical significance, *p* < 0.05, *p* < 0.01. Abbreviations: ICU, intensive care unit; MV, mechanical ventilation; SD, standard deviation.

**Table 6 jpm-13-01497-t006:** The logistic regression model for in-hospital mortality. Exp (B) is equivalent to the odds ratio (OR), a measure of a relationship’s strength between the predictor and the binary outcome.

**Model**: Deceased ~ Age + COVID-19 + N comorbidities + Ground glass opacities over 50% Controlling for: GenderM + ActiveSmoker + pandemicWave (categorical) + extension of SP–SPM (categorical)				
	**Predictor**	**B ± Std. err**	***p*-Value**	**Exp (B)** (95% CI)
	Age	0.042 ± 0.02	0.039 *	1.043 (1.002–1.085)
	COVID-19	1.797 ± 0.629	0.004 **	6.032 (1.757–20.712)
	N comorbidities	0.572 ± 0.269	0.033 *	1.772 (1.046–3.001)
	Ground glass opacities over 50%	1.739 ± 0.808	0.031 *	5.694 (1.169–27.746)
	Nagelkerke R-square = 0.524			

Abbreviations: B ± Std. err, coefficient of regression ± standard error; CI, confidence interval. *, ** Statistical significance, *p* < 0.05, *p* < 0.01.

**Table 7 jpm-13-01497-t007:** The logistic regression model for required intensive care. Exp (B) is equivalent to the odds ratio (OR), a measure of a relationship’s strength between the predictor and the binary outcome.

**Model:** ICU ~ Age + COVID-19 + N comorbidities + Ground glass opacities over 50% Controlling for: GenderM + ActiveSmoker + pandemicWave (categorical) + extension of SP–SPM (categorical)				
	**Predictor**	**B ± Std. err**	***p*-Value**	**Exp (B)** (95% CI)
	Age	0.011 ± 0.016	0.511	1.011 (0.979–1.043)
	COVID-19	1.901 ± 0.613	0.002 **	6.693 (2.013–22.261)
	N comorbidities	0.504 ± 0.246	0.041 *	1.656 (1.022–2.683)
	Ground glass opacities over 50%	1.570 ± 0.736	0.033 *	4.807 (1.137–20.326)
	Nagelkerke R-square = 0.419			

Abbreviations: B ± Std. err, coefficient of regression ± standard error; CI, confidence interval; ICU, intensive care unit. *, ** Statistical significance, *p* < 0.05, *p* < 0.01.

**Table 8 jpm-13-01497-t008:** The logistic regression model for required mechanical ventilation. Exp (B) is equivalent to the odds ratio (OR), a measure of a relationship’s strength between the predictor and the binary outcome.

**Model:** MV ~ Age + COVID-19 + N comorbidities + Ground glass opacities over 50% Controlling for: GenderM + ActiveSmoker + pandemicWave (categorical) + extension of SP–SPM (categorical)				
	**Predictor**	**B ± Std. err**	***p*-Value**	**Exp (B)** (95% CI)
	Age	0.012 ± 0.016	0.454	1.012 (0.981–1.044)
	COVID-19	1.249 ± 0.622	0.045 *	3.488 (1.031–11.801)
	N comorbidities	0.519 ± 0.254	0.041 *	1.681 (1.022–2.766)
	Ground glass opacities over 50%	1.928 ± 0.712	0.007 *	6.876 (1.704–27.735)
	Nagelkerke R-square = 0.373			

Abbreviations: B ± Std. err, coefficient of regression ± standard error; CI, confidence interval; MV, mechanical ventilation. * Statistical significance, *p* < 0.05.

## Data Availability

The datasets are not publicly available, but de-identified data may be provided upon request from Adina Maria Marza.

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
