# Peer review of "Non-Ventilated Patients with Spontaneous Pneumothorax or Pneumomediastinum Associated with COVID-19: Three-Year Debriefing across Five Pandemic Waves"

_jpm, 2023, doi:10.3390/jpm13101497_

Round 1

Reviewer 1 Report

The current study titled “Non-Ventilated Patients with Spontaneous Pneumothorax or Pneumomediastinum Associated with COVID-19: Three-Year Debriefing across Five Pandemic Waves” Ref: 2643793, deals with an important subject. Although mild number of patients was considered, the author(s) did his best to extract the most important results attained. Minor revisions are needed considering the following items.

- The previously published studies of this subject should be highlighted in the introduction section with their fruitful results.

- The conclusion section should be revised and supported by the observations and results of the current study.

Author Response

Dear Reviewer,

We appreciate your feedback and thoughtful recommendations on how to enhance the material. We are confident that the process significantly improved our manuscript. Please find below the responses to the questions and issues you’ve raised.

In addition, we submit the revised manuscript with yellow highlights (so the corresponding changes are more easily identified).

1.”The previously published studies of this subject should be highlighted in the introduction section with their fruitful results.”

Answer:

We included two paragraphs within the "Introduction" section (highlighted in yellow), where we added three recent studies addressing the same topic, along with their results.

2.”The conclusion section should be revised and supported by the observations and results of the current study.”

Answer:

We revised the ”Conclusion” section to more effectively convey our study's findings.

We value the time you have taken to provide us with your input and we would be glad to receive any additional feedback you may have.

Thank you on behalf of the authors,

Faithfully yours,

Alina Petrica, Md, PhD

Reviewer 2 Report

1.Introduction, more detailed previous literature reviews of spontaneous pneumothorax and pneumodiastinum was suggested in the preface, and add some description of insufficient previous studies.

2.Discussion, discussion of the strengths, limitations and practical implications of this study is missing.

3.Discussion, the discussion needs to go beyond just stating consistency with previous research and expanding on possible causes, etc., and overall, it is not deep enough.

4.Discussion, the authors can give recommendations related to how to decrease in-hospital mortality with spontaneous pneumothorax and pneumomediastinumm during the COVID-19 pandemic based on the results of this study.

5.Tables should be produced in accordance with the norms of textual publication, e.g. a three-line table should be used.

6.Conclusion, the authors may consider streamlining this section to allow for more focus.

7.Overall, I believe the manuscript may benefit from a thorough review of the language.

English needs a little editing.

Author Response

Dear Reviewer,

We appreciate your feedback and thoughtful recommendations on how to enhance the material. We are confident that the process significantly improved our manuscript. Please find below the responses to the questions and issues you’ve raised. In addition, we submit the revised manuscript with yellow highlights (so the corresponding changes are more easily identified).

”1.Introduction, more detailed previous literature reviews of spontaneous pneumothorax and pneumodiastinum was suggested in the preface, and add some description of insufficient previous studies.”

Answer:

We included two paragraphs within the "Introduction" section (highlighted in yellow), where we added three recent studies addressing the same topic, along with their results.

”2.Discussion, discussion of the strengths, limitations and practical implications of this study is missing.”

Answer:

Following your suggestions, we made extensive revisions to the entire Disscusion section and added a previously absent limitations paragraph. Additionally, we addressed the study's strengths and explored certain practical implications.

” 3.Discussion, the discussion needs to go beyond just stating consistency with previous research and expanding on possible causes, etc., and overall, it is not deep enough.”

Answer:

As you suggested, we made extensive revisions throughout the Disscusion section. We restructured the discussions to create a more coherent narrative, and we corroborated our results both with the potential causes and with the results from previous research.

” 4.Discussion, the authors can give recommendations related to how to decrease in-hospital mortality with spontaneous pneumothorax and pneumomediastinumm during the COVID-19 pandemic based on the results of this study.”

Answer:

In the final paragraph of the Discussion, we included recommendations that we deem significant, drawing from our experience as physicians who have cared for COVID-19 patients throughout the three-year pandemic. These recommendations pertain to the management of patients with SP-SPM and COVID-19, and they are based on the study's findings.

” 5.Tables should be produced in accordance with the norms of textual publication, e.g. a three-line table should be used.”

Answer:

All tables were thoroughly revised. Given the denser nature of the information presented in the tables, particularly since they encompass data spanning five waves of the pandemic, for reasons of readability and maintaining the correct meaning of the results, we chose to keep visible lines between the rows of the tables. Nonetheless, if it is considered mandatory for the journal’s format to adopt a different approach, we will comply with the requirements.

” 6.Conclusion, the authors may consider streamlining this section to allow for more focus.”

Answer

We revised the Conclusion  section to more effectively convey our study's findings.

” 7.Overall, I believe the manuscript may benefit from a thorough review of the language.”

Answer

We conducted a thorough English review and indeed identified errors, which we subsequently corrected.

We value the time you have taken to provide us with your input and we would be glad to receive any additional feedback you may have.

Thank you on behalf of the authors,

Faithfully yours,

Alina Petrica, Md, PhD

Round 2

Reviewer 2 Report

There are no modification suggestions and agree to publish the article.